# Analysis of cervical and breast cancer screening behavior and its influencing factors among urban and rural women in Beijing

Yanqing Gao[1☯], Guoxin Liang[1☯], Chun Chang[2], Feng Pan[1], Fumin Huang[3], Litong Gong[1]*, Liyu Huang[4]*

1 Daxing Center for Disease Control and Prevention, Beijing, China, 2 Department of Social Medicine and Health Education, Peking University School of Public Health, Beijing, China, 3 Faculty Development Center, Chinese Academy of Medical Sciences & Peking Union Medical College, Beijing, China, 4 Beijing Center for Disease Prevention and Control, Beijing, China

☯ These authors contributed equally to this work.
* gonglitong2024@126.com (LG); liyu_huang0526@126.com (LH)

**Data Availability Statement:** All relevant data are within the paper and its Supporting information files.

## Abstract

### Objective

To clarify the screening behavior and influencing factors of females with breast cancer and cervical cancer in suburban areas and to provide a scientific basis for the subsequent implementation of targeted health education, intervention measures and the formulation of relevant policies.

### Methods

This study used a multi-stage stratified random sampling method to select 4, 000 women in urban and rural areas of Beijing to analyze their behavior, basic situation, and influencing factors regarding cervical and breast cancer screening.

### Results

The sample size of the final included valid analysis was 3861 people, and the screening rate was 27.25% for cervical cancer, 20.64% for breast cancer, 30.46% for at least one screening and 17.43% for both cervical cancer and breast cancer screening. The rate of four screening conditions was greater in urban areas ($P_{Cervical\ cancer\ screening}$ = 31.1%, $P_{Breast\ cancer\ screening}$ = 22.0%, $P_{At\ least\ one}$ = 33.9%, $P_{Both\ cancers\ were\ screened}$ = 19.1%) than in rural areas ($P_{Cervical\ cancer\ screening}$ = 22.6%, $P_{Breast\ cancer\ screening}$ = 19.0%, $P_{At\ least\ one}$ = 26.2%, $P_{Both\ cancers\ were\ screened}$ = 15.4%) and was greater with medical insurance ($P_{Cervical\ cancer\ screening}$ = 28.7%, $P_{Breast\ cancer\ screening}$ = 21.7%, $P_{At\ least\ one}$ = 32.0%, $P_{Both\ cancers\ were\ screened}$ = 18.5%) than without medical insurance ($P_{Cervical\ cancer\ screening}$ = 12.8%, $P_{Breast\ cancer\ screening}$ = 10.3%, $P_{At\ least\ one}$ = 15.6%, $P_{Both\ cancers\ were\ screened}$ = 7.5%). The highest percentage of the four screening conditions was found in the 45–59-year-old group ($P_{Cervical\ cancer\ screening}$ = 36.0%, $P_{Breast\ cancer\ screening}$ = 29.8%, $P_{At\ least\ one}$ = 39.5%, $P_{Both\ cancers\ were\ screened}$ = 26.4%). The rate of cervical cancer screening behavior increased with increasing education

**Funding:** The author(s) received no specific funding for this work.

**Competing interests:** The authors have declared that no competing interests exist.

level and family per capita monthly income, and the highest percentage of respondents had a college education or above ($P_{Cervical\ cancer\ screening}$ = 35.2%, $P_{Breast\ cancer\ screening}$ = 23.6%, $P_{At\ least\ one}$ = 38.2%, $P_{Both\ cancers\ were\ screened}$ = 20.6%), as did the percentage of families whose per capita monthly income was above 15, 000 yuan ($P_{Cervical\ cancer\ screening}$ = 34.7%, $P_{Breast\ cancer\ screening}$ = 27.3%, $P_{At\ least\ one}$ = 38.3%, $P_{Both\ cancers\ were\ screened}$ = 23.6%). Multivariate analysis revealed that an age range of 45 to 59 years ($P_{All\ four\ screening\ conditions\ were\ obtained}$<0.001), an education level of junior high school, a high school ($P_{All\ four\ screening\ conditions\ were\ obtained}$<0.001), a college education or above ($P_{All\ four\ screening\ conditions\ were\ obtained}$<0.001), a marital status of a spouse ($P_{All\ four\ screening\ conditions\ were\ obtained}$<0.001), a divorce status ($P_{All\ four\ screening\ conditions\ were\ obtained}$<0.001) or a widowhood status ($P_{All\ four\ screening\ conditions\ were\ obtained}$<0.001), and a medical insurance status ($P_{All\ four\ screening\ conditions\ were\ obtained}$<0.001) were positively correlated with the percentages of the four screening behaviors.

## Conclusion

The level of "two- cancer" screening behavior of suburban residents in Beijing still warrants improvement, and precision nutrition and health communication and intervention should be carried out continuously for rural residents, individuals under age 45, unmarried individuals, individuals with a primary school education and below, and people without medical insurance.

## Introduction

Women's health reflects the progress and development of society, and it is an important health indicator that measures the population's health quality, quality of life, and level of civilization in a country [1]. According to the latest data released by the International Agency for Research on Cancer (IARC), nearly 2.3 million new cases of breast cancer were reported in 185 countries, and nearly 670, 000 women died from breast cancer in 2022, which ranks first in both incidence and mortality among women worldwide. There were 661, 000 new cases of cervical cancer and 348, 000 related deaths, ranking fourth in terms of global cancer incidence and mortality among women [2]. In 2022, China had 151, 000 new cases of cervical cancer, accounting for 22.84% of the world's new cases. In the same year, China had 357, 000 new cases of breast cancer, accounting for 15.52% of the world's new cases [3]. Among women aged 25–49 years in China, breast cancer and cervical cancer are the malignant tumor types with the highest mortality rates [4]. Cervical cancer and breast cancer have become major public health concerns in current society. Changes in vaginal discharge and breast lumps, among other clinical symptoms, can indicate early stages of cervical cancer and breast cancer [5]. Early detection, diagnosis and treatment of cervical cancer and breast cancer can help improve the survival rate and quality of life of patients [6–9]. "Two-cancer" screening refers to cervical cancer and breast cancer screening. "Two-cancer" screening is a screening program for women without symptoms through effective, simple, and economical examination measures to identify precancerous lesions in the early stage and provide treatment. The goal of cancer screening is to reduce the mortality rate of patients with cervical cancer and breast cancer [10]. The Chinese government has included breast cancer and cervical cancer screening in basic public health service projects and has launched a "two-cancer screening" project for eligible

women aged 35 to 64 years in urban and rural areas across the country to enhance screening efforts for women's cancers [11]. "Two-cancer" screening is a public welfare activity carried out in China to protect women's health, and it is pivotal for the prevention and treatment of breast cancer and cervical cancer [12]. Due to the influence of various factors, the participation of adult women in breast cancer screening in China has been unsatisfactory [9, 13, 14]. An investigation and analysis were conducted to clarify the screening behavior and influencing factors of the "two-cancer" in suburban women, to carry out targeted health education and intervention measures, and to provide a scientific basis for the formulation of related policies.

## Objects and methods

### Objects

Residents of Beijing aged 15 to 80 years The exclusion criteria were as follows: mentally disabled patients, senile dementia patients, psychotic patients in the attack period (patients in the fluctuation period need to have their family members present) and other uncooperative patients.

## Methods

### Sampling method

This survey was a cross-sectional study, and a multi-stage stratified random sampling method was used to select 4, 000 female residents in 20 towns and rural areas of Beijing. A total of 3, 861 effective questionnaires were collected, with an effective rate of 96.53%, including 2, 122 for urban women and 1, 749 for rural women.

### Survey methods

The questionnaire for this survey was designed with reference to the "Adult Tobacco Use Survey Questionnaire in Chinese Cities", the "Survey Questionnaire on Chronic Diseases and Risk Factors in China" and the "Dietary Nutrition Survey Questionnaire". The questionnaire included social demographic information, daily living habits, community guidance and knowledge of chronic disease prevention and control. Physical measurements included height, weight, waist circumference and blood pressure. Height, weight and blood pressure were measured by uniformly trained investigators using metrology-certified instruments designated by a national program (TZG height sitting height meter, BMI HD-390 electronic weighing scale, Omron HBP1300 electronic blood pressure monitor). The readings were accurate to 0.1 cm, 0.1 kg and 1 mmHg, respectively, and were measured centrally using standard methods.

   Institutional Review Board Statement: The study was conducted in accordance with the "Declaration of Helsinki", "Operational Guidelines for Ethics Committees That Review Biomedical Research", "International Ethical Guidelines for Biomedical Research Involving Human Subjects" and other international ethical guidelines. And this study was approved in writing by the Biomedical Ethics Committee of Peking University (PU IRB) with the approval code IRB00001052-22148 (Date of approval: December 12, 2022). Participants were recruited from December 30, 2022 to May 15, 2023, on-site questionnaires and physical tests were conducted. All study participants signed the informed consent forms, and we obtained consent from parents or guardians of the minors included in the study.

### Indicator definition

Behavior possession rate = the number of respondents who hold a certain behavior/the total number of respondents ×100%. Urban areas refer to places where the land use rights of the

houses where residents' household registration addresses are located belong to state-owned land use rights. Rural areas refer to places where the land use rights of the house on the registered residence belong to collective land use rights. Self-rated health statusrefers to the subjective evaluation of one's own health. In this study, the health status is divided into three categories: bad, average and good.

## Quality control

All the staff at the monitoring points had unified training, methods, standards, equipment, data input and cleaning. At the same time, a quality control group was set up, with responsibilities and tasks stipulated, to be responsible for the inspection and supervision of on-site work.

## Statistical methods

In this study, SPSS 19.0 software was used to clean up the database and carry out general statistical descriptions. The differences between the behavioral holding rates of people with different characteristics were tested by the $\chi^2$ test, and the influencing factors were analyzed by logistic regression analysis. The test level was $\alpha = 0.05$.

# Results

## General situation

A total of 3, 861 effective analysis samples were included in this paper. Among them, there were 2, 099 people aged <45 (54.4%), 975 people aged 45~ (25.3%), and 787 people aged 60~ (20.4%). A total of 2, 112 people were urban residents (54.7%), and 1, 749 people were suburban residents (45.3%). The majority of residents were of Han nationality (3, 647, 95.2%), 438 people had a primary or lower education level (11.3%), 1, 650 people had junior high school or high school education (42.7%), 1, 773 people had college or higher education (45.9%), 3, 102 people were married (80.3%), 1, 568 people had a family monthly income of 2, 500~ (40.6%), 3, 051 people had medical insurance (90.7%), and 2, 451 people self-evaluated their health status as good (63.5%). Table 1.

## Behavior holding

**The holding rate of each behavior.** Among the 3, 861 residents, 1, 052 women participated in cervical cancer screening, for a screening rate of 27.25%. The rate of cervical cancer screening behavior was greater in urban areas (31.1%) than in rural areas (22.6%), was greater in those with insurance (28.7%) than in those without insurance (12.8%), and was greatest among those aged 45–54 (36.0%). A total of 31.4% and 35.5% of the respondents were spouse and divorced, respectively. In general, 29.6% of the participants had self-assessed health status. The rate of cervical cancer screening behavior increased with higher education level and per capita monthly income, reaching the highest levels among those with a college degree or above, as did those with a per capita monthly income of 15, 000 yuan or above (35.2% and 34.7%, respectively). The above differences were all statistically significant (P < 0.05) Table 2.

Among the 3861 residents, 797 women participated in breast cancer screening, for a screening rate of 20.64%. The breast cancer screening rate was greater in urban areas (22.0%) than in rural areas (19.0%) and in those with medical insurance (21.7%) than in those without medical insurance (10.3%). A total of 23.9% and 19.4% of those with a spouse or divorced, respectively, had good self-rated health status. The breast cancer screening rate increased with increasing education level and family per capita monthly income, with a college degree or above and a

**Table 1. Basic information of urban and rural women in Beijing City [n (%)].**

| Group | age | | | Total |
|---|---|---|---|---|
| | <45 | 45~ | 60~ | |
| **Area** | | | | |
| Town | 1183 (56.4) | 479 (49.1) | 450 (57.2) | 2112 (54.7) |
| Rural area | 916 (43.6) | 496 (50.9) | 337 (42.8) | 1749 (45.3) |
| **The Han nationality** | | | | |
| No | 1970 (93.9) | 943 (96.7) | 761 (96.7) | 3674 (95.2) |
| Yes | 129 (6.1) | 32 (3.3) | 26 (3.3) | 187 (4.8) |
| **Degree of education** | | | | |
| Primary school and below | 39 (1.9) | 127 (13.0) | 272 (34.6) | 438 (11.3) |
| Junior high school and senior high school | 599 (28.5) | 605 (62.1) | 446 (56.7) | 1650 (42.7) |
| College degree or above | 1461 (69.6) | 243 (24.9) | 69 (8.8) | 1773 (45.9) |
| **Marriage** | | | | |
| Unmarried | 526 (25.1) | 7 (0.7) | 3 (0.4) | 536 (13.9) |
| Have a spouse | 1517 (72.3) | 899 (92.2) | 686 (87.2) | 3102 (80.3) |
| Dissociation | 49 (2.3) | 41 (4.2) | 17 (2.2) | 107 (2.8) |
| Bereft of one's spouse | 7 (0.3) | 28 (2.9) | 81 (10.3) | 116 (3.0) |
| **Per capita monthly household income** | | | | |
| Less than 2500 | 256 (12.2) | 263 (27.0) | 212 (26.9) | 731 (18.9) |
| 2500~ | 817 (38.9) | 385 (39.5) | 366 (46.5) | 1568 (40.6) |
| 6000~ | 614 (29.3) | 181 (18.6) | 107 (13.6) | 902 (23.4) |
| 15000~ | 412 (19.6) | 146 (15.0) | 102 (13.0) | 660 (17.1) |
| **Medical insurance** | | | | |
| No | 176 (8.4) | 104 (10.7) | 80 (10.2) | 360 (9.3) |
| Yes | 1923 (91.6) | 871 (89.3) | 707 (89.8) | 3501 (90.7) |
| **Self-rated health status** | | | | |
| Bad | 46 (2.2) | 25 (2.6) | 23 (2.9) | 94 (2.4) |
| Average | 715 (34.1) | 355 (36.4) | 246 (31.3) | 1316 (34.1) |
| Good | 1338 (63.7) | 595 (61.0) | 518 (65.8) | 2451 (63.5) |
| Total | 2099 (54.4) | 975 (25.3) | 787 (20.4) | 3861 (100) |

family per capita monthly income above 15, 000 reaching the highest values of 23.6% and 27.3%, respectively. The above differences were statistically significant (P<0.05). Table 2.

Among the 3861 residents, 1176 women participated in at least one type of cervical cancer or breast cancer screening, for a screening rate of 30.46%. The percentage of individuals with at least one type of cervical cancer and breast cancer screening behavior was greater in urban areas (33.9%) than in rural areas (26.2%) and greater in those with medical insurance (32.0%) than in those without medical insurance (15.6%). The 45-year-old group (39.5%) was the most likely to be 45 years old, with relatively higher percentages of 34.8% and 37.4% for spouses and divorced people, respectively, and 33.7% for self-rated health status in general. The percentage of women with cervical cancer screening increased with increasing education level and family per capita monthly income, with the highest percentage of those with a college degree or above and a family per capita monthly income above 15000 accounting for 38.2% and 38.3%, respectively. The above differences were statistically significant (P<0.05). See Table 2.

Among the 3861 residents, 673 women participated in both cervical cancer and breast cancer screening, resulting in a screening rate of 17.43%. Both cervical cancer and breast cancer screening rates were greater in urban areas (19.1%) than in rural areas (15.4%) and were

**Table 2. The rate of cervical cancer and breast cancer screening behavior in Beijing [n (%)].**

| group | number of people | Cervical cancer screening | Breast cancer screening | At least 1 | Both have |
|---|---|---|---|---|---|
| Area | | | | | |
| Town | 2112 | 657 (31.1) | 464 (22.0) | 717 (33.9) | 404 (19.1) |
| Rural area | 1749 | 395 (22.6) | 333 (19.0) | 459 (26.2) | 269 (15.4) |
| $\chi^2$ | | 35.064 | 5.015 | 26.817 | 9.341 |
| P | | 0.000 | 0.025 | 0.000 | 0.002 |
| Age | | | | | |
| <45 | 2099 | 600 (28.6) | 402 (19.2) | 657 (31.3) | 345 (16.4) |
| 45~ | 975 | 351 (36.0) | 291 (29.8) | 385 (39.5) | 257 (26.4) |
| 60~ | 787 | 101 (12.8) | 104 (13.2) | 134 (17.0) | 71 (9.0) |
| $\chi^2_{trend}$ | | 41.395 | 1.847 | 29.447 | 5.844 |
| P | | 0.000 | 0.174 | 0.000 | 0.016 |
| The Han nationality | | | | | |
| No | 187 | 46 (24.6) | 30 (16.0) | 47 (25.1) | 29 (15.5) |
| Yes | 3674 | 1006 (27.4) | 767 (20.9) | 1129 (30.7) | 644 (17.5) |
| $\chi^2$ | | 0.695 | 2.538 | 2.631 | 0.505 |
| P | | 0.404 | 0.111 | 0.105 | 0.477 |
| Degree of education | | | | | |
| Primary school and below | 438 | 34 (7.8) | 38 (8.7) | 48 (11.0) | 24 (5.5) |
| Junior high school and senior high school | 1650 | 394 (23.9) | 341 (20.7) | 451 (27.3) | 284 (17.2) |
| College degree or above | 1773 | 624 (35.2) | 418 (23.6) | 677 (38.2) | 365 (20.6) |
| $\chi^2_{trend}$ | | 147.579 | 38.042 | 133.425 | 46.571 |
| P | | 0.000 | 0.000 | 0.000 | 0.000 |
| Marriage | | | | | |
| Unmarried | 536 | 27 (5.0) | 16 (3.0) | 35 (6.5) | 8 (1.5) |
| Have a spouse | 3102 | 973 (31.4) | 742 (23.9) | 1080 (34.8) | 635 (20.5) |
| Dissociation | 107 | 38 (35.5) | 23 (21.5) | 40 (37.4) | 21 (19.6) |
| Bereft of one's spouse | 116 | 14 (12.1) | 16 (13.8) | 21 (18.1) | 9 (7.8) |
| $\chi^2_{trend}$ | | 39.488 | 35.310 | 49.265 | 27.034 |
| P | | 0.000 | 0.000 | 0.000 | 0.000 |
| Per capita monthly household income | | | | | |
| Less than 2500 | 731 | 167 (22.8) | 161 (22.0) | 202 (27.6) | 123 (17.2) |
| 2500~ | 1568 | 362 (23.1) | 260 (16.6) | 399 (25.4) | 223 (14.2) |
| 6000~ | 902 | 294 (32.6). | 196 (21.7) | 322 (35.7) | 168 (18.6) |
| 15000~ | 660 | 229 (34.7) | 180 (27.3) | 253 (38.3) | 156 (23.6) |
| $\chi^2_{trend}$ | | 43.585 | 12.558 | 37.708 | 16.639 |
| P | | 0.000 | 0.000 | 0.000 | 0.000 |
| Medical insurance | | | | | |
| No | 360 | 46 (12.8) | 37 (10.3) | 56 (15.6) | 27 (7.5) |
| Yes | 3501 | 1006 (28.7) | 760 (21.7) | 1120 (32.0) | 646 (18.5) |
| $\chi^2$ | | 41.930 | 26.035 | 41.629 | 27.240 |
| P | | 0.000 | 0.000 | 0.000 | 0.000 |
| Self-rated health status | | | | | |
| Bad | 94 | 25 (26.6) | 21 (22.3) | 30 (31.9) | 16 (17.0) |
| Average | 1316 | 389 (29.6) | 300 (22.8) | 443 (33.7) | 246 (18.7) |
| Good | 2451 | 638 (26.0) | 476 (19.4) | 703 (28.7) | 411 (16.8) |
| $\chi^2_{trend}$ | | 3.890 | 5.488 | 8.604 | 1.576 |
| P | | 0.049 | 0.019 | 0.003 | 0.209 |
| Total | 3861 | 1052 | 797 | 1176 | 673 |

greater in those with medical insurance (18.5%) than in those without medical insurance (7.5%). The 45-year-old group had the greatest percentage of patients in the 45-year-old group (26.4%), with relatively higher percentages of 20.5% and 19.6% for spouses and divorced people, respectively. The rate of cervical cancer screening behavior increased with increasing education level and family per capita monthly income, with the highest percentage of individuals holding a college degree and above and a family per capita monthly income above 15000 (20.6% and 23.6%, respectively). The above differences were statistically significant (P<0.05). See Table 2.

**Influencing factors of screening behavior.**   Multivariate logistic regression model analysis revealed that cervical cancer screening, breast cancer screening, and screening of at least one or both cancers were the dependent variables, and region, age, ethnicity, education level, marital status, family monthly income per capita, medical insurance and self-health status were the independent variables. The results of all four screening methods revealed the following: the age group of 45~ years old, the education level in middle school, high school and junior college or above, marital status as a spouse, divorced, bereft of one's spouse, and having Medicare were positively associated with the holding rate of the four screening behaviors. However, the family of per capita monthly income of 2500~ yuan were negatively associated with each of the four screening conditions. See Table 3.

## Discussion

In this study, 1052 women out of 3861 residents aged 15 to 80 years participated in cervical cancer screening, and the screening rate was 27.25%. A global statistical study of cervical cancer screening coverage showed that an estimated 1.6 billion (67%) of 2.3 billion women aged 20–70 years had never been screened for cervical cancer. In 2014–2019, the worldwide coverage of cervical cancer screening was 32% among women aged 30 to 49 years, that the coverage rate of cervical cancer screening of High income Countries was 77%, and Lower-middle-income Countries was 28%. The coverage rate of cervical cancer screening in women aged 35 to 49 years was 33% in China [15]. The cervical cancer screening rate for women aged 15 to 69 years in Guangdong Province was 15.5% [16]. In the suburbs of Beijing, the cervical cancer screening rate was lower than that in Tianjin (37.98%) [17]. This value was lower than that of 44.6% in Shanghai [18]. A total of 797 women participated in breast cancer screening, for a screening rate of 20.64%, which was lower than the 22.5% rate reported by national surveillance data [13]. The breast cancer screening rate for women aged 15~69 years in Guangdong Province was 15.1% [16]. The breast cancer screening rate for women aged 35~64 years in Sichuan Province was 31.7% [19]. There were 1176 women who had participated in at least one cervical or breast cancer screening, for a screening rate of 30.46%. There were 673 women involved in both cervical cancer and breast cancer screening, for a screening rate of 17.43%. A total of 33.5% of residents aged 35~64 years in Dongcheng District of Beijing participated in free cervical cancer and breast cancer screening [20]. Therefore, according to the data reported in existing studies, although the screening rates are different between different provinces and cities in China, they are generally low compared with developed countries, and there is still a long way to go to promote screening for these two cancers.

The results of univariate analysis in this study show that the holding rate of four screening situations in urban residents is greater than that in rural residents, and the differences in the holding rate of the other three screening situations are statistically significant except for breast cancer screening. The results of cervical cancer screening are the same as those of Tao Hua and others [21], but the results of breast cancer screening are opposite to those of Bao Hailing [13] and others, which may be due to the economic and cultural levels in different regions.

**Table 3. Multivariate logistic regression model of "two-cancer" screening behavior in Beijing.**

| group | Cervical cancer screening | | Breast cancer screening | | Two cancers had at least one type | | Both cancers were screened | |
|---|---|---|---|---|---|---|---|---|
| | OR (95%CI) | P price | OR (95%CI) | P price | OR (95%CI) | P price | OR (95%CI) | P price |
| Area | | | | | | | | |
| Town* | 1.00 | | 1.00 | | 1.00 | | 1.00 | |
| Rural area | 1.134 (0.949~1.357) | 0.167 | 1 (0.825~1.212) | 0.999 | 1.068 (0.899~1.269) | 0.456 | 1.077 (0.876~1.323) | 0.481 |
| Age | | | | | | | | |
| <45* | 1.00 | | 1.00 | | 1.0 | | 1.0 | |
| 45~ | 1.619 (1.333~1.965) | 0.000 | 1.854 (1.51~2.277) | 0.000 | 1.608 (1.33~1.945) | 0.000 | 1.928 (1.555~2.39) | 0.000 |
| 60~ | 0.579(0.441~0.761) | 0.000 | 0.877 (0.661~1.162) | 0.36 | 0.687 (0.533~0.885) | 0.004 | 0.719 (0.524~0.986) | 0.041 |
| The Han nationality | | | | | | | | |
| No* | 1.00 | | 1.00 | | 1.00 | | 1.00 | |
| Yes | 0.866(0.599~1.252) | 0.445 | 0.771 (0.509~1.168) | 0.22 | 0.748 (0.52~1.075) | 0.117 | 0.931 (0.608~1.425) | 0.742 |
| Degree of education | | | | | | | | |
| Primary school and below* | 1.00 | | 1.00 | | 1.00 | | 1.00 | |
| Junior high school and senior high school | 3.211(2.19~4.707) | 0.000 | 2.751 (1.901~3.983) | 0.000 | 2.828 (2.025~3.951) | 0.000 | 3.305 (2.115~5.165) | 0.000 |
| College degree or above | 6.914(4.541~10.525) | 0.000 | 4.809 (3.166~7.305) | 0.000 | 6.228 (4.276~9.072) | 0.000 | 5.689 (3.482~9.294) | 0.000 |
| Marriage | | | | | | | | |
| Unmarried* | 1.00 | | 1.00 | | | | | |
| Have a spouse | 11.296(7.537~16.93) | 0.000 | 10.817 (6.471~18.081) | 0.000 | 9.911 (6.9~14.236) | 0.000 | 18.214 (8.947~37.079) | 0.000 |
| Dissociation | 11.834 (6.645~21.075) | 0.000 | 8.579 (4.275~17.217) | 0.000 | 9.794 (5.683~16.878) | 0.000 | 15.272 (6.449~36.166) | 0.000 |
| Bereft of one's spouse | 5.68(2.759~11.693) | 0.000 | 7.797 (3.646~16.674) | 0.000 | 6.716 (3.587~12.573) | 0.000 | 8.798 (3.218~24.053) | 0.000 |
| Per capita monthly household income | | | | | | | | |
| Less than 2500* | 1.00 | | 1.00 | | 1.00 | | 1.00 | |
| 2500~ | 0.765(0.606~0.965) | 0.024 | 0.598 (0.47~0.761) | 0.000 | 0.686 (0.55~0.856) | 0.001 | 0.658 (0.507~0.855) | 0.002 |
| 6000~ | 0.863(0.662~1.126) | 0.278 | 0.666 (0.504~0.88) | 0.004 | 0.805 (0.624~1.039) | 0.095 | 0.698 (0.516~0.942) | 0.019 |
| 15000~ | 1.353(1.042~1.758) | 0.023 | 1.141 (0.874~1.49) | 0.332 | 1.258 (0.979~1.615) | 0.072 | 1.241 (0.931~1.655) | 0.14 |
| Medical insurance | | | | | | | | |
| No* | 1.00 | | 1.00 | | 1.00 | | 1.00 | |
| Yes | 2.006(1.428~2.817) | 0.000 | 1.983 (1.373~2.865) | 0.000 | 1.886 (1.377~2.584) | 0.000 | 2.258 (1.483~3.438) | 0.000 |
| Self-rated health status | | | | | | | | |
| Bad* | 1.00 | | 1.00 | | 1.00 | | 1.00 | |
| Average | 1.127(0.675~1.88) | 0.648 | 1.022 (0.602~1.736) | 0.935 | 1.038 (0.638~1.688) | 0.88 | 1.136 (0.633~2.039) | 0.67 |
| Good | 1.032(0.623~1.712) | 0.902 | 0.907 (0.538~1.53) | 0.715 | 0.888 (0.549~1.435) | 0.628 | 1.099 (0.617~1.958) | 0.749 |

Note:

* is the reference group.

The holding rates of the four screening situations in the 45~ age group were greater than those in the age groups less than 45 years and greater than 60 years, and the lowest was in the age group greater than 60 years. With the exception of breast cancer screening, there were statistically significant differences in the behavioral holding rates of the other three screening situations. This finding is similar to the research results of Tao Hua, Bao Heling and many others [21–24]. The higher the educational level of the patient, the greater the holding rate of the four screening conditions, and a college education or above has the

highest holding rate. Among the residents with different marital statuses, those with a spouse and those who are divorced have relatively higher screening rates for the four behaviors, which may indicate that people with sexual experience are more concerned about their own health and that married women have more family support from their husbands and children, as well as more sources of health knowledge. This finding is the same as the research results of Tao Hua, Bao Heling, Shao Ying [13, 21, 25] et al. The higher the household monthly income is, the greater the rate of adherence to the four screening behaviors among residents, which is consistent with the results of multiple studies [13, 23, 26, 27], possibly because high-income groups have better access to medical resources and good screening availability. On the other hand, low-income women, regardless of how low the screening cost is, will always be a factor that needs to be considered. Residents with health insurance have higher rates of screening behaviors for the four screening conditions, consistent with the findings of studies by Tao Hua, Bao Hailing, Wang B, et al. [13, 21, 28]. Women with lower incomes will have greater expenses for breast cancer screening. Even if they participate in free screening programs, they will worry about being diagnosed with breast cancer. Without insurance reimbursement for medical expenses, the family burden will greatly increase, which leads to a decreased willingness of women to undergo screening. Having health insurance is an important factor in accessing health services, and women without insurance have a worse perception of their health [29]. The highest rates of screening behaviors are found in residents with an average self-rated health status. Except for those who had both cancer screenings, the differences in self-rated health status were statistically significant for the other three types of screening behaviors, possibly because this group of people is more able to objectively assess their own health levels, they are not blindly confident, and they do not shy away from seeking medical treatment. The results of the multivariate study show that the following factors are positively correlated with the rate of having the four screening behaviors: being aged 45~, having a middle school or high school education, having a college degree or higher, being married, divorced or widowed and having health insurance.

## Conclusion

In conclusion, the level of screening for "two-cancer" among suburban residents in Beijing still warrants improvement. Health Science Popularization and intervention for accurate "two-cancer" screening should be carried out continuously for residents who are rural, under 45 years old, are unmarried, have a primary school education or below and lack medical insurance. Health behavior theories related to cervical and breast cancer screening, such as health belief theory, planned behavior theory, and self-efficacy theory, can be used to improve the compliance of residents and strengthen awareness of "two-cancer" prevention control as well as screening behavior in this group of women. At the same time, enhancing the supply capacity of screening services and expanding the scope of screening can improve the coverage of "two-cancer" screening services [30]. We must increase awareness and screening rates for breast and cervical cancer among residents through different means to reduce the incidence and mortality rates of these two cancers.

## Supporting information

**S1 Dataset. Primary data.**
(XLS)

## Acknowledgments

We thank the Daxing District Center for Disease Prevention and Control, and related staff for their support and assistance during the investigation.

## Author Contributions

**Conceptualization:** Yanqing Gao, Guoxin Liang, Litong Gong, Liyu Huang.

**Data curation:** Yanqing Gao, Guoxin Liang, Litong Gong, Liyu Huang.

**Formal analysis:** Yanqing Gao, Guoxin Liang, Fumin Huang, Litong Gong, Liyu Huang.

**Funding acquisition:** Yanqing Gao, Guoxin Liang, Litong Gong.

**Investigation:** Yanqing Gao, Guoxin Liang, Feng Pan, Fumin Huang, Litong Gong.

**Methodology:** Yanqing Gao, Guoxin Liang, Feng Pan, Fumin Huang, Litong Gong, Liyu Huang.

**Project administration:** Yanqing Gao, Guoxin Liang, Chun Chang, Feng Pan, Litong Gong.

**Resources:** Yanqing Gao, Guoxin Liang, Chun Chang, Feng Pan, Fumin Huang, Litong Gong.

**Software:** Litong Gong.

**Supervision:** Yanqing Gao, Guoxin Liang, Chun Chang, Litong Gong.

**Validation:** Litong Gong.

**Visualization:** Litong Gong.

**Writing – original draft:** Yanqing Gao, Guoxin Liang, Litong Gong.

**Writing – review & editing:** Litong Gong, Liyu Huang.

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
