## [Decision Letter · Decision Letter 0]

23 Oct 2024

PONE-D-24-23150

Analysis of cervical and breast cancer screening behavior and its influencing factors among urban and rural women in Beijing

PLOS ONE

Dear Dr. Huang,

Thank you for submitting your manuscript to PLOS ONE. After careful consideration, we feel that it has merit but does not fully meet PLOS ONE’s publication criteria as it currently stands. Therefore, we invite you to submit a revised version of the manuscript that addresses the points raised during the review process.

We look forward to receiving your revised manuscript.

Kind regards,

Pengpeng Ye

Academic Editor

PLOS ONE

Journal Requirements:

file:///C:/Users/e754871/Desktop/PRTC/PRTC%202024/October%202024/October%2023,%202024/Journal%20requirements:%20%0b%0bWhen%20submitting%20your%20revision,%20we%20need%20you%20to%20address%20these%20additional%20requirements.%20%0b%0b1.%20Please%20ensure%20that%20your%20manuscript%20meets%20PLOS%20ONE's%20style%20requirements,%20including%20those%20for%20file%20naming.%20The%20PLOS%20ONE%20style%20templates%20can%20be%20found%20at%20https:/journals.plos.org/plosone/s/file%3fid=wjVg/PLOSOne_formatting_sample_main_body.pdf%20and When submitting your revision, we need you to address these additional requirements.

2. We note that you have indicated that there are restrictions to data sharing for this study. PLOS only allows data to be available upon request if there are legal or ethical restrictions on sharing data publicly. For more information on unacceptable data access restrictions, please see http://journals.plos.org/plosone/s/data-availability#loc-unacceptable-data-access-restrictions. Before we proceed with your manuscript, please address the following prompts: a) If there are ethical or legal restrictions on sharing a de-identified data set, please explain them in detail (e.g., data contain potentially identifying or sensitive patient information, data are owned by a third-party organization, etc.) and who has imposed them (e.g., a Research Ethics Committee or Institutional Review Board, etc.). Please also provide contact information for a data access committee, ethics committee, or other institutional body to which data requests may be sent. b) If there are no restrictions, please upload the minimal anonymized data set necessary to replicate your study findings to a stable, public repository and provide us with the relevant URLs, DOIs, or accession numbers. For a list of recommended repositories, please see https://journals.plos.org/plosone/s/recommended-repositories. You also have the option of uploading the data as Supporting Information files, but we would recommend depositing data directly to a data repository if possible. We will update your Data Availability statement on your behalf to reflect the information you provide.

4. We are unable to open your Supporting Information file “primary data .sav”. Please kindly revise as necessary and re-upload.

Reviewers' comments:

Reviewer's Responses to Questions

**Comments to the Author**

1. Is the manuscript technically sound, and do the data support the conclusions?

Reviewer #1: Partly

Reviewer #2: Yes

2. Has the statistical analysis been performed appropriately and rigorously? 

Reviewer #1: I Don't Know

Reviewer #2: Yes

3. Have the authors made all data underlying the findings in their manuscript fully available?

Reviewer #1: Yes

Reviewer #2: Yes

4. Is the manuscript presented in an intelligible fashion and written in standard English?

Reviewer #1: Yes

Reviewer #2: Yes

5. Review Comments to the Author

Reviewer #1: This survey was a cross-sectional study about the factors influencing screening for breast cancer and cervical cancer in Beijing. However, the results have not been presented precisely and the conclusions are not related to the results. Here, I have some questions for the authors.

1. Could you please explain what is unique about your study? Representative due to the Capital city of China?

2. For the discussion section, the authors only compared the findings with several cities that are not well-known from my perspective. Please explain in more detail the clinical meaning of the comparisons.

3. The results section is just a simple list of data, lacking highly refined information and conclusions.

4. In table 1, what is the definition of the 'Self-rated health status'?

5. Currently, there are four main types of basic health insurance in China. What is the kind of 'Hospitalization insurance' in this study?

Reviewer #2: Current study seems well established. The methodology is clearly defined, including a multi-stage stratified random sampling method and appropriate statistical analyses using SPSS. The sample size is sufficiently large to support the conclusions drawn, and the study provides detailed comparisons between urban and rural areas as well as different demographic factors, such as age, education level, and medical insurance status.

6. PLOS authors have the option to publish the peer review history of their article (what does this mean?). If published, this will include your full peer review and any attached files.

Reviewer #1: No

Reviewer #2: **Yes: **JIYU LI

---

## [Author Response · Author response to Decision Letter 0]

17 Nov 2024

Dear Editors and Reviewers:

Thank you for your letter and for the reviewers’ comments.I have modified the format of the article accordingly. The data of this study were uploaded as supporting documents. Available to all. The following is the response to the questions raised by the reviewers.

1.Could you please explain what is unique about your study? Representative due to the Capital city of China?

This study mainly discusses the screening behavior and influencing factors of women with two cancers in urban and suburban areas of Beijing. The sample size is relatively large, which represents the current situation of two cancer screening in the capital of China to some extent.

2. For the discussion section, the authors only compared the findings with several cities that are not well-known from my perspective. Please explain in more detail the clinical meaning of the comparisons.

Discussion section compares the results of the survey in Beijing and different provinces in China, can reflect the differences in different regions within China. Through comparison, the gap can be found, and the authenticity of the data can be reflected to a certain extent. In the mean time, global data are added to the article, and the gap is found through comparison at home and abroad, which provides a basis for better screening of the two cancers and promoting the implementation of the screening policy of the two cancers.

3. The results section is just a simple list of data, lacking highly refined information and conclusions.

The data are described in the results section of this paper, and the results are refined and discussed in the discussion section.

4. In table 1, what is the definition of the 'Self-rated health status'?

Self-rated health status is the subjective evaluation of one's own health. This study divides the health status into three categories: poor, general and good, which has been explained in the index definition section of the article.

5. Currently, there are four main types of basic health insurance in China. What is the kind of 'Hospitalization insurance' in this study?

Many thanks for proposing and discovering the problem. In this study, medical insurance refers to basic medical insurance, which is a social insurance system established to compensate workers for the economic losses caused by disease risk. The “Hospitalization insurance” in the table is the “Medical insurance” in the text, which has been made in the table.

---

## [Decision Letter · Decision Letter 1]

18 Dec 2024

Analysis of cervical and breast cancer screening behavior and its influencing factors among urban and rural women in Beijing

PONE-D-24-23150R1

Dear Dr. Huang,

We’re pleased to inform you that your manuscript has been judged scientifically suitable for publication and will be formally accepted for publication once it meets all outstanding technical requirements.

Kind regards,

Pengpeng Ye

Academic Editor

PLOS ONE

Additional Editor Comments (optional):

Reviewers' comments:

Reviewer's Responses to Questions

**Comments to the Author**

1. If the authors have adequately addressed your comments raised in a previous round of review and you feel that this manuscript is now acceptable for publication, you may indicate that here to bypass the “Comments to the Author” section, enter your conflict of interest statement in the “Confidential to Editor” section, and submit your "Accept" recommendation.

Reviewer #2: All comments have been addressed

2. Is the manuscript technically sound, and do the data support the conclusions?

Reviewer #2: Yes

3. Has the statistical analysis been performed appropriately and rigorously? 

Reviewer #2: Yes

4. Have the authors made all data underlying the findings in their manuscript fully available?

Reviewer #2: Yes

5. Is the manuscript presented in an intelligible fashion and written in standard English?

Reviewer #2: Yes

6. Review Comments to the Author

Reviewer #2: Given the significant improvements in addressing reviewers' concerns, refining the analysis, and strengthening the discussion, it is recommended that the manuscript be accepted for publication. The authors have demonstrated responsiveness to feedback and made substantial revisions that enhance the study's clarity, relevance, and contribution to the field.

7. PLOS authors have the option to publish the peer review history of their article (what does this mean?). If published, this will include your full peer review and any attached files.

Reviewer #2: No

---

## [Editor Report · Acceptance letter]

14 Jan 2025

PONE-D-24-23150R1 

PLOS ONE

Dear Dr. Huang, 

I'm pleased to inform you that your manuscript has been deemed suitable for publication in PLOS ONE. Congratulations! Your manuscript is now being handed over to our production team.

Kind regards, 

on behalf of

Dr. Pengpeng Ye 

Academic Editor

PLOS ONE